# POLICY OPTIMIZATION EMERGES FROM NOISY REPRESENTATION LEARNING

## ABSTRACT

Nervous systems learn representations of the world and policies to act within it. We present a framework that uses reward-dependent noise to facilitate policy optimization in representation learning networks. These networks balance extracting normative features and task-relevant information to solve tasks. Moreover, their representation changes reproduce several experimentally observed shifts in the neural code during task learning. Our framework presents a biologically plausible mechanism for emergent policy optimization amid evidence that representation learning plays a vital role in governing neural dynamics. Code is available at: NeuralThermalOptimization.

## 1    INTRODUCTION

Nervous systems learn representations of the world. They configure their weights to extract certain information from the environment: for instance, the presence of objects or their temporal dynamics (Li & DiCarlo (2008); de Vries & Wurm (2023)). This learning is driven by synaptic plasticity (Markram et al. (1997); Bi & Poo (1998); Dan & Poo (2004)). Representation learning objectives provide a normative framework for proposing plasticity rules (Lipshutz et al. (2023); van Hateren (1992); Pehlevan et al. (2015)). Mounting evidence suggests that these objectives largely determine synaptic plasticity. Rules derived from these objectives emulate experimentally observed spike-rate dependent plasticity (Pehlevan & Chklovskii (2014); Pehlevan et al. (2017); Qin et al. (2023); Sengupta et al. (2018); Halvagal & Zenke (2023); Tang et al. (2024); Millidge et al. (2024)). At the same time, these rules reproduce population-level changes to the neural code seen under a variety of experimental conditions (Raju et al. (2024); Qin et al. (2023); Halvagal & Zenke (2023)). However, organisms do not only build representations of the environment; they also use them to optimize goal-directed behavior. The mechanisms that underlie this behavioral learning remain unclear.

This article introduces **NE**ural **T**hermal **O**ptimization (NETO), a framework for how policy optimization can emerge from noisy representation learning. NETO treats the brain's intrinsic noise as a feature rather than a bug. It hypothesizes that the noise in the nervous system is reward-dependent, decreasing in magnitude when the nervous system's weight configuration yields a rewarding policy. This modulation, akin to thermal cooling, alters the attractive points in weight space, encouraging the network to optimize its policy while learning a representation objective.

We illustrate the framework by considering an agent whose one-layer network operates under a simple, linear representation learning objective. We analyze the agent's learning process in two tasks. In the first, the features defined by the agent's objective capture all task-relevant information. Through analytical arguments and simulations, we show that the agent is guaranteed to learn an optimal policy by diffusively searching the space of equivalent representations. In the second task, the agent needs to learn features not captured by its representation learning objective to perform well. We show that NETO balances normative and task-relevant feature learning to solve the task. These results suggest that NETO is a plausible theory for policy learning in nervous systems.

## 2    RELATED WORK

A key question in theoretical neuroscience is how organisms learn to interact in environments. Early work showed that reward-modulated spike-timing dependent plasticity (STDP) rules can implement

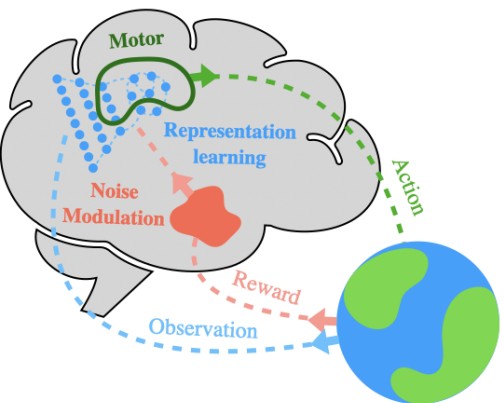

**Figure 1:** *A schematic of the Neural Thermal Optimization framework. A representation learning network (blue) receives observations from the environment and extracts features. A motor function (green) determines an action given the network activities, defining the agent's policy. A modulatory system (red) tunes the noise in the representation learning network according to its estimate of the policy reward.*

reinforcement learning (Florian (2007); Legenstein et al. (2008)). This model gained traction because its structure is consistent with experimental observations. Organisms with vastly different nervous systems, from the *Caenorhabditis elegans* worm to humans, possess neuromodulatory systems (Shine et al. (2019); Qin & Wheeler (2007); Alcedo & Prahlad (2020)). Moreover, their STDP rules are consistent with those measured across organisms. However, artificial networks operating under this framework can only solve simple tasks like discriminating different spike trains.

Representation learning may offer a path forward. In artificial agents, representation learning objectives greatly improve task-learning capabilities (Jaderberg et al. (2016); Ha & Schmidhuber (2018)). For biological models, promising work showed that spiking neural networks that learn policies from a world model solve simple Atari games (Capone & Paolucci (2024)). However, this work was based on rules with little experimental validation and necessitated separate modules for representation and policy learning.

We also use representation learning as a foundation but do not impose an additional module for policy learning. Instead, we return to the idea that a neuromodulatory system acts on a network operating under experimentally validated plasticity rules. However, we allow the modulation to tune the nervous system's intrinsic noise. The use of noise as a computational tool is not new (Kappel et al. (2017); Maass (2014); Li et al. (2024)). But, to our knowledge, the key idea that noise can act as an effective weight temperature, "cooling" the network in rewarding configurations, has not yet been explored.

## 3 FRAMEWORK

### 3.1 THE MARKOV DECISION PROCESS

A natural framework for an organism interacting in an environment is the Markov Decision Process (MDP). We define an MDP with the tuple $(\mathcal{S}, \mathcal{A}, P, r, \gamma)$. The environment's dynamics are defined by the state space $\mathcal{S}$, the agent's action space $\mathcal{A}$, and $P(s'|s, a)$, the probability of transitioning from a state $s$ to a state $s'$ given an action $a$. Upon this transition, the agent receives a reward specified by the function $r(s, a)$.

The agent acts according to a policy $\pi : \mathcal{S} \times \mathcal{A} \rightarrow [0, 1]$ which describes the distribution of actions that it could take in a state $s$. We denote this distribution $\pi(a|s)$. Its goal is to learn a policy that maximizes its reward over a finite time horizon $T$. If we let the subscript $t$ denote the timestep, the reward is given by $R(\pi) = \mathbb{E}[\sum_{t=1}^{T} \gamma^t r(s_t, a_t)]$, where $a_t \sim \pi(a|s_t)$ per the definition of the policy, and $\gamma \in (0, 1]$ is the temporal discount factor.

## 3.2 Neural Thermal Optimization

Here, we introduce Neural Thermal Optimization (NETO). In NETO agents, a single noisy neural network learns representations and controls actions, while a neuromodulatory system tunes the noise. As such, NETO has three components: a representation learning network, a motor function that defines the agent's policy, and reward-dependent noise controlled by a neuromodulatory system. We illustrate each in Figure 1, and introduce them below, along with the example that we consider throughout this work.

### 3.2.1 A representation learning objective governs the network dynamics

Consistent with biology, we consider an online agent, meaning that it learns through interacting continuously with its environment. We begin by describing its representation learning network. Let the neural network be parameterized by activities $\boldsymbol{x} \in \mathbb{R}^N$ and weights $\theta$. This network can have any architecture. The network's goal is to extract features from the environment. It builds representations of the MDP, in principle capturing aspects of the distribution of $s$ or the dynamics of the state transitions defined by $P$.

The network extracts features by optimizing a representation learning objective $\mathcal{L}(\boldsymbol{x}, \theta, s)$. We refer to the features that it learns as "normative" because $\mathcal{L}$ specifies how the network weights should change to extract them. Because the agent is online, its activities and weights respond together to each state $s$. These responses are noisy, and are given by the stochastic differential equations

$$d\boldsymbol{x} = -\eta_x \frac{\partial \mathcal{L}}{\partial \boldsymbol{x}} dt + \sigma_x dB \tag{1}$$

$$d\theta = -\eta_\theta \frac{\partial \mathcal{L}}{\partial \theta} dt + \sigma_\theta dB \tag{2}$$

where the $dB$ are independent standard Wiener processes in the necessary number of dimensions, $\eta_x$ and $\eta_\theta$ are the weight and activity learning rates, and $\sigma_x$ and $\sigma_\theta$ are the diffusion tensors of the stochastic process. Typically, $\eta_x >> \eta_\theta$ such that the activity timescale is much faster than the weight timescale. The activities respond to the state $s$, and then the weights change according to these responses.

We can make this discussion more concrete by introducing our simple example agent. As we will see, this agent exhibits many important features of NETO while remaining easily interpretable and analytically tractable. We let our agent's representation learning network optimize the Similarity Matching (SM) objective, introduced in (Pehlevan et al. (2015))[1]. The SM objective encourages the network to project its inputs onto their principal subspace. Nicely, this objective bestows the network with Hebbian weight updates, which emphasizes that our model is biologically plausible.

Consider a one-layer network with activities $\boldsymbol{x} \in \mathbb{R}^2$ and weights $\theta = (\boldsymbol{W}_0, \boldsymbol{W}_1)$. The first weight matrix $\boldsymbol{W}_0$ is a feedforward matrix, and $\boldsymbol{W}_1$ is a matrix of lateral inhibitory connections[2]. Suppose that the agent receives $D$-dimensional state vectors $\boldsymbol{s} \in \mathbb{R}^D$ from the environment. Then $\boldsymbol{W}_0 \in \mathbb{R}^{D \times 2}$ and $\boldsymbol{W}_1 \in \mathbb{R}^{2 \times 2}$. Previous work (Pehlevan et al. (2017)) showed that the network satisfies the SM objective if it optimizes

$$\mathcal{L}(\boldsymbol{x}, \boldsymbol{W}_0, \boldsymbol{W}_1, \boldsymbol{s}) = \mathrm{Tr}(\boldsymbol{W}_0^T \boldsymbol{W}_0) - \frac{1}{2}\mathrm{Tr}(\boldsymbol{W}_1^T \boldsymbol{W}_1) - 2\boldsymbol{x}^T \boldsymbol{W}_0 \boldsymbol{s} + \boldsymbol{x}^T \boldsymbol{W}_1 \boldsymbol{x} \tag{3}$$

giving it the dynamics

$$d\boldsymbol{x} = \eta_x(\boldsymbol{W}_0 \boldsymbol{s} - \boldsymbol{W}_1 \boldsymbol{x})dt \tag{4}$$

$$d\boldsymbol{W}_0 = \eta_\theta(\boldsymbol{x}\boldsymbol{s}^T - \boldsymbol{W}_0)dt + \sqrt{\eta_\theta}\sigma_\theta dB \tag{5}$$

---

[1]The original, offline SM objective is defined as $\|\boldsymbol{S}^T\boldsymbol{S} - \boldsymbol{X}^T\boldsymbol{X}\|^2$, where $\boldsymbol{S} = [\boldsymbol{s}_1 \ \boldsymbol{s}_2 \ ...]$ is a matrix of inputs and $\boldsymbol{X} = [\boldsymbol{x}_1 \ \boldsymbol{x}_2 \ ...]$ is a matrix of outputs. We will consider the online formulation (3).

[2]$W_1$ must be symmetric and positive-definite.

$$dW_1 = \eta_\theta(\boldsymbol{x}\boldsymbol{x}^T - \boldsymbol{W}_1)dt + \sqrt{\eta_\theta}\sigma_\theta dB \tag{6}$$

For simplicity, we set the activity noise to zero to make the network's activity dynamics deterministic. We will see soon that this also makes the agent's policy deterministic, which will simplify future analysis. We also let $\sigma_\theta$ be a scalar rather than a tensor. At each timestep of the MDP, the activities evolve under the differential equation (4) until they converge, followed by a discrete weight update.

### 3.2.2 A MOTOR FUNCTION DEFINES THE AGENT'S POLICY

Under NETO, the same neural network learns representations and controls actions. We now introduce the motor function, which defines the agent's policy. Let $M : \mathbb{R}^N \to \mathcal{A}$ be a motor function that maps the network activities to an action. We assume that $M$ is independent of time[3]. Biologically, we can imagine that this motor function arises due to projections from the neural activities $\boldsymbol{x}$ to a motor control system or from movements of the muscles directly controlled by $\boldsymbol{x}$.

This motor function defines the agent's policy, the distribution of possible actions $a$ that it could take in response to a state $s$. In the general case of noisy activity dynamics, a distribution $f_\theta(\boldsymbol{x}|s)$ describes the possible network responses to a state $s$. The policy induced by the motor function is given by the distribution of the actions associated with these network responses

$$\pi_\theta(a|s) = \int f_\theta(\boldsymbol{x}|s)\,\delta(a - M(\boldsymbol{x}))\,d\boldsymbol{x} \tag{7}$$

where $\delta(.)$ is the Dirac delta function. This relation holds in general for both discrete and continuous actions.

Now we return to our example. For simplicity, our agent has the discrete action space $\mathcal{A} = \{-1, 1\}$. We define the motor function $M(\mathbf{x}) = \text{sign}(\boldsymbol{v}^T\boldsymbol{x})$, where $\boldsymbol{v}^T = (-1, 1)$. This choice defines a linear classifier in activity (representation) space. Notice that our agent's policy is deterministic; we set its activity noise to zero, so it maps a state $s$ deterministically to an activity pattern, which in turn defines an action through $M$.

### 3.2.3 REWARD-DEPENDENT NOISE FACILITATES POLICY OPTIMIZATION

The final and key component of NETO is the reward-dependent noise. We define the agent's neuromodulatory system so that it shapes the representation learning network's weight diffusion tensor according to an estimate of the policy reward $R$

$$\sigma_\theta = \sigma_\theta(R) \tag{8}$$

where the eigenvalues of $\sigma_\theta(R)$ are monotonically decreasing functions of $R$. Intuitively, this reward-dependent noise "cools" the system when the agent's policy is rewarding. More formally, it balances optimization of $\mathcal{L}$ and $R$ by changing the system's stochastically attractive points in weight space. In Appendix A.3, we explain how activity noise can also play this role. It is not unreasonable to conceptualize $R$ as a regularizer on $\mathcal{L}$.

In our example agent, we define a modulatory system that estimates $R$ and controls the noise through $\eta_\theta = \eta_\theta(R)$, as the noise is proportional to $\sqrt{\eta_\theta}$ [4]. We discuss the biological plausibility of this choice in Appendix A.4. In particular, we let

$$\eta_\theta(R) = 0.01e^{-\beta R}. \tag{9}$$

for $\beta > 0$. This choice of monotonically decreasing function is arbitrary.

---

[3]Time independence is an approximation that is not necessary but will simplify the discussion.

[4]Since all weights have the same learning rate, we implicitly restrict the diffusion tensor to a multiple of the identity here. This restriction likely hinders policy learning efficiency, as discussed in the Limitations section.

While this work focuses on introducing NETO for policy optimization in representation learning networks, we feel it is important to note that the framework is actually more general. We discuss this generality in Appendix A.1.

# 4 RESULTS

## 4.1 A DIFFUSIVE SEARCH FACILITATES POLICY OPTIMIZATION WHEN $\mathcal{L}$ CAPTURES THE TASK-RELEVANT INFORMATION

An ideal representation learning objective captures enough information to solve a variety of tasks. We begin by analyzing the agent's learning process when this is this case. We construct a contextual bandit task that the agent can solve with the features that $\mathcal{L}$ encourages it to extract. We then show that the agent is guaranteed to maximize reward as time goes to infinity [5]. In the process, we illustrate that a diffusive search through the space of equivalent representations facilitates policy learning: The network parameters converge to the solution space of $\mathcal{L}$ and explore it using reward-dependent noise.

### 4.1.1 THE CONTEXTUAL BANDIT TASK

The contextual bandit task is a special case of the Markov Decision Process where the state transitions are independent of the agent's action. Let our contextual bandit task have states $s \in \mathbb{R}^{10}$. The states are drawn from two Gaussian clusters in 10 dimensions. In particular, we define them as $s = ky$, where $y \sim \mathcal{N}(\mu, C)$, and $k$ is a random sign. The largest eigenvalue of the covariance matrix $C$ is 0.1, and the rest are 0.05. The mean vector $\mu$ is the second eigenvector of $C$ and has magnitude 2. We also define the agent's action space as $\mathcal{A} = \{1, -1\}$, and let $r(s, a) = \mathbf{1}_{(a=k)}$. The agent must learn to take the action corresponding to the random sign. It is possible to learn this task because the Gaussian clusters are linearly separable. Figure 2 illustrates the setup.

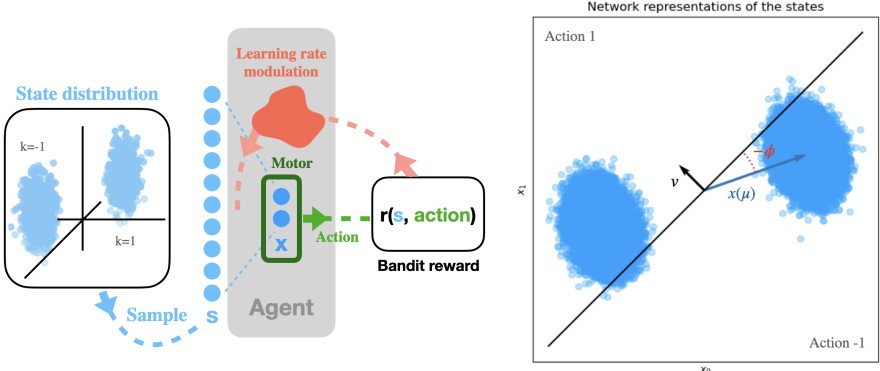

**Figure 2:** *(Left) Schematic of the agent in the contextual bandit task. The agent receives a state $s$ (light blue), and its activities respond, resulting in an action. The neuromodulatory system receives the reward associated with this action and adjusts the weight learning rate. The weights then update according to $\mathcal{L}$, and the task continues. The agent's goal is to take the action corresponding to the random sign $k$. (Right) Illustration of all relevant variables in the network's representation space, where the axes denote the activity of its two neurons. Royal blue points show the network representations of the states $s$, denoted $x(s)$. The vector $v$ defines a classifier that maps these representations to the action $1$ or $-1$. Finally, $\phi$, the angle between $v$ and $x(\mu)$, tracks the rotational diffusion of the network responses through policy learning.*

### 4.1.2 $\mathcal{L}$ CAPTURES THE TASK-RELEVANT FEATURE AND HAS A SOLUTION SPACE

We aim to show that the agent learns an optimal policy, even though the network that controls its actions only explicitly optimizes $\mathcal{L}$. We begin by demonstrating that (1) the network's objective captures the information necessary to solve the task, and (2) that this objective has a solution space: the network representations are free to rotate. We will use these properties to equate policy learning to

---

[5]In practice, this occurs much more quickly.

a diffusive search through the solution space of $\mathcal{L}$. Finally, we will compute the limiting distribution of this search with reward-dependent noise. This process will show us that the agent preferentially resides in rewarding orientations.

Recall that $\mathcal{L}$ encourages the network to project the states $s$ onto their principal subspace. We chose $\mu$ so that its only nonzero component is in the principal subspace of $s$. This means that the network needs to extract the features defined by $\mathcal{L}$ to solve the task. After it does so, its activities contain the information needed to pick the action associated with the random sign $k$. However, the agent does not necessarily use this information correctly; the stimulus-to-action mapping defined by $M$ is arbitrary.

The agent learns a rewarding policy because its representation learning network's weights diffuse through the solution space of $\mathcal{L}$. Representation learning objectives often have solution spaces because they are degenerate or the network is overparameterized. Here, the objective is degenerate. We begin by finding its solution space. This was done in previous work (Qin et al. (2023)), but we review it here because it is necessary for the rest of our discussion.

There are infinite ways to project the states $s$ onto their principal subspace. In particular, any transformation of the form

$$W_0 \to W_0' = UW_0 \tag{10}$$

$$W_1 \to W_1' = UW_1U^T \tag{11}$$

which yields

$$x \to x' = Ux \tag{12}$$

where $U^TU = I$ leaves $\mathcal{L}$ invariant. In other words, any rotation of the network representations is still a projection onto the principal subspace. Note that, to arrive at equation (12), we assumed that the activity dynamics (4) converge before a weight update.

We can now describe the solution space of $\mathcal{L}$ in terms of the network activities. Suppose that the network projects its inputs onto their principal subspace (such that its weights are in the solution space of $\mathcal{L}$). Let $x(s)$ denote its response to the state $s$. Then the space of all responses to this state that satisfy $\mathcal{L}$ are

$$\mathcal{H}_s = \{Ux(s) : U \in SO(2)\} \tag{13}$$

where $SO(2)$ is the special orthogonal group in two dimensions[6]. This space is a circle.

### 4.1.3 WITH NOISE, THE AGENT EXPLORES THE SOLUTION SPACE OF $\mathcal{L}$

When the weight noise is reward-independent, the network representations converge to and explore their respective $\mathcal{H}_s$. Random weight transformations of the form (10-11), with $U$ as an infinitesimal rotation, generate this noisy exploration. The representations undergo rotational diffusion. We can describe the system's configuration in the solution space by tracking the evolution of $\phi$, the angle between $x(\mu)$ and $v$. By analyzing noisy perturbations to this configuration, previous work showed that the diffusion coefficient characterizing the process is linear in $\eta_\theta$ (Qin et al. (2023)). So, its evolution is given by

$$d\phi \propto \sqrt{\eta_\theta}dB \tag{14}$$

where we work up to proportionality because we are only interested in its limiting distribution. Since all representations transform together, this single parameter $\phi$ describes the network configuration during policy learning.

---

[6]Technically, $U \in O(2)$, but this extension is not important to our discussion because we will not consider reflections.

#### 4.1.4 REWARD-DEPENDENT NOISE BIASES THE NETWORK TOWARDS REWARDING POLICIES

Now that we understand the network's solution space exploration, we can analyze it with reward-dependent noise modulated by $\eta_\theta(R)$. In a general task with reward-dependent noise, we cannot guarantee that the network responses converge to $\mathcal{H}_s$. Reward-dependent noise can stabilize points outside of the solution space. We will discuss this in greater detail later.

However, Figure 3 shows that the principal subspace error rapidly decreases and remains near optimal throughout this task. Here, all representations $\boldsymbol{x}(\boldsymbol{s})$ are approximately contained to their $\mathcal{H}_s$. Intuitively, this occurs because these representations contain the information needed to solve the task. Reasonable policies exist within the solution space of $\mathcal{L}$. We can therefore analyze the reward-dependent rotational diffusion of the representation $\boldsymbol{x}(\boldsymbol{s})$ through $\mathcal{H}_s$.

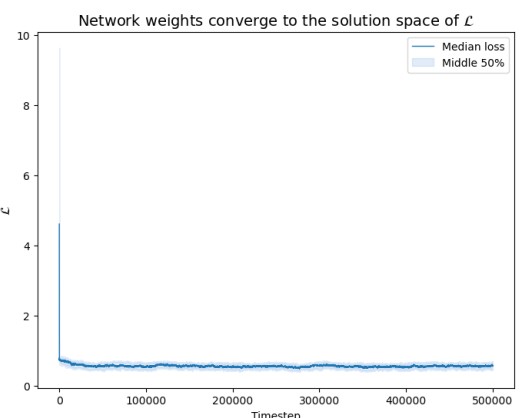

**Figure 3:** *The principal subspace error over time, quantified by $\frac{\|\boldsymbol{F}^T\boldsymbol{F}-\boldsymbol{U}\boldsymbol{U}^T\|_F}{\|\boldsymbol{U}\boldsymbol{U}^T\|_F}$. The columns of $\boldsymbol{U}$ are the top two eigenvectors of the covariance matrix $\boldsymbol{C}$, and $\boldsymbol{F} = \boldsymbol{W}_1^{-1}\boldsymbol{W}_0$ is the projection performed by the network after the convergence of (4). The operation $\|.\|_F$ denotes Frobenius norm.*

Reward-dependent rotational diffusion is equivalent to angle-dependent diffusion. This equivalence arises from the fact that $R$ is a function of $\phi$. One can see graphically in Figure 2, but as an example, notice that $M[\boldsymbol{x}(\boldsymbol{\mu}|\phi = \frac{\pi}{2})] = 1$, whereas $M[\boldsymbol{x}(\boldsymbol{\mu}|\phi = -\frac{\pi}{2})] = -1$. Different angles correspond to different policies.

If we find $\eta_\theta(\phi)$, we can express the rotational diffusion in a self-contained manner. We show in Appendix A.5 that $R(\phi)$ is approximately

$$R(\phi) \approx \Phi\left(\pm\frac{\|\boldsymbol{\mu}\|}{\sqrt{\lambda_1}}\tan\phi\right) \tag{15}$$

where $\Phi$ is the standard normal CDF, and $\lambda_1$ is the largest eigenvalue of $C$. We take the negative sign when $\phi \in [\frac{\pi}{2}, \frac{3\pi}{2}]$ and the positive sign otherwise. We can therefore write $\eta_\theta(\phi) = \eta_\theta(R(\phi))$, which is given by

$$\eta_\theta(\phi) \approx \exp\left[-\beta\Phi\left(\pm\frac{\|\boldsymbol{\mu}\|}{\sqrt{\lambda_1}}\tan\phi\right)\right] \tag{16}$$

Plugging (16) into (14) gives the equation describing the rotational diffusion of the network response to $\boldsymbol{\mu}$ through the agent's learning process. The same expression holds for all network responses. Under the Ito prescription, this is equivalent to the Fokker-Planck equation (up to proportionality)

$$\frac{\partial p(\phi, t)}{\partial t} \propto \frac{\partial^2}{\partial\phi^2}\left[\eta_\theta(\phi)p(\phi, t)\right]. \tag{17}$$

where $p(\phi, t)$ is the distribution of $\phi$. Its stationary distribution $p(\phi)$ is therefore

$$p(\phi) \propto \frac{1}{\eta_\theta(\phi)} \tag{18}$$

Finally, we can use $p(\phi)$ to find the expected reward received by an agent. In this limit, it is given by

$$\mathbb{E}[R] = \int_0^{2\pi} R(\phi)p(\phi)\,d\phi \tag{19}$$

which we compute numerically. In Figure 4, we see that as $\beta$ increases, $\mathbb{E}[R] \to 1$. With strong modulation, the agent is guaranteed to learn a policy that maximizes reward.

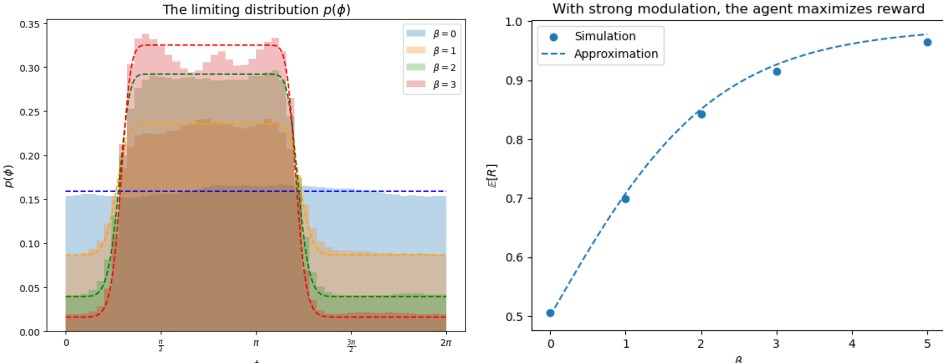

**Figure 4:** *(Left) The limiting distribution of $\phi$. The dashed lines denote our approximations for different $\beta$, and the histograms show the simulation data. With $\beta = 0$ (no noise modulation), $\phi$ is uniformly distributed in the solution space of $\mathcal{L}$. With larger $\beta$, the agent becomes biased towards rewarding angles. Data generated from 100 seeds of 500,000 timesteps. (Right) The expected reward received by the agent in the limit that time goes to infinity. The agent is guaranteed to maximize reward for large $\beta$ (stronger modulation). Our approximation agrees with simulations.*

Notice that policy optimization emerges by modulating a noisy, reward-independent objective. The agent first learns to extract normative features (the principal subspace projection). Then, it uses this information to optimize its policy (by exploring the informatically equivalent rotational configurations). In the next example, we will see that NETO is not restricted to this sequential learning: It can balance normative feature extraction with policy optimization by performing both simultaneously.

## 4.2 NETO BALANCES NORMATIVE REPRESENTATION AND POLICY LEARNING TO SOLVE TASKS

The contextual bandit task gives the impression that NETO agents learn representations and policies in separate phases. However, in general, NETO couples representation and policy learning. This coupling allows agents to balance normative and task-dependent representation learning, extracting features that $\mathcal{L}$ does not capture to improve task performance.

The contextual bandit example illustrates that we can consider the $R$-dependent noise as $\theta$-dependent. This equivalence arises because the same network learns representations and controls actions. Recent work in the context of stochastic gradient descent found that the weight diffusion tensor $\sigma_\theta(\theta)$ and the loss $\mathcal{L}(\theta)$ jointly determine the attractive points in weight configuration space (Chen et al. (2024)). This fact gives the network the freedom to extract information that is not optimal with respect to $\mathcal{L}$ if it helps the agent perform well in the task.

We demonstrate this principle with Cart Pole. Cart Pole is a classical control problem with a four-dimensional observation space. It is simple enough for our one-layer, linear network to solve. The goal of Cart Pole is to keep a pole upright on a cart while preventing it from falling off a frictionless track. The agent can push the cart right or left at each timestep. Importantly, the agent cannot learn an optimal policy for Cart Pole if its weights project the state observations onto their two-dimensional principal subspace, shown empirically in Appendix A.6. Therefore, the agent cannot solve the task with only normative features; it must learn to adjust its representations so that they contain information that is suboptimal with respect to $\mathcal{L}$.

Indeed, we find that solving the task requires the agent's weights to stabilize at representation learning losses that are orders of magnitude larger than in the contextual bandit task, as seen in Figure 5 (left). The agent learns to extract features not specified by $\mathcal{L}$. Note that these features are not arbitrary. They are still partially determined by $\mathcal{L}$; the loss settles to be many orders of magnitude lower than where it began.

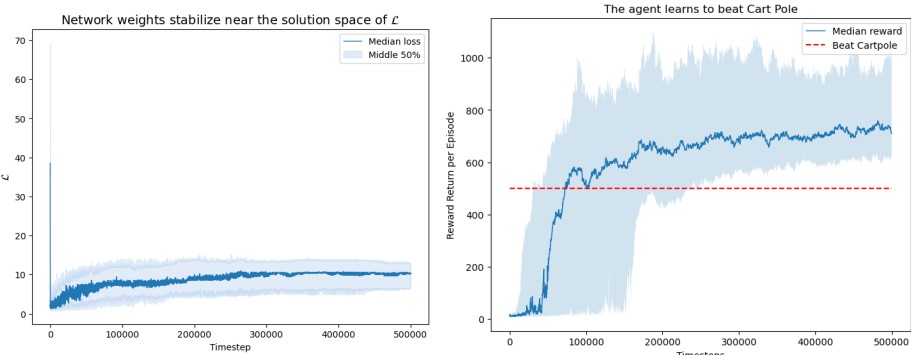

**Figure 5:** *(Left) The principal subspace loss over time. The loss decreases rapidly through initial feature learning but settles an order of magnitude larger than in the contextual bandit task. Reward-dependent noise stabilized the weights in a suboptimal configuration to extract task-relevant features. (Right) The agent learns to beat Cart Pole. Median and middle 50% percentile over 100 seeds plotted through 500,000 timesteps of continuous interaction.*

This simple example illustrates a general principle: NETO agents balance normative and task-relevant representation learning to facilitate policy optimization. In general, the NETO framework is most likely to extract task-relevant features if they exist near the solution space of $\mathcal{L}$. Policy learning acts as a regularizer on weights that are otherwise selected to extract normative features defined by $\mathcal{L}$. Representation and policy learning are interdependent, emerging together from the same system.

## 5 DISCUSSION

In this article, we present the hypothesis that policy optimization emerges from the noisy learning of representation objectives in nervous systems. We explore the consequences of this idea in a linear, one-layer network. Through simulations and analytical arguments, we show that this network learns the optimal policy in tasks that it can reasonably be expected to solve. Each task serves to provide intuition for its learning process. When $\mathcal{L}$ captures all task-relevant features, the network performs a diffusive search through its solution space to discover an optimal policy. The agent can also modify its representations to capture task-relevant features, even if the resulting representations are slightly suboptimal with respect to $\mathcal{L}$.

### 5.1 CONSISTENCY WITH EXPERIMENTAL OBSERVATIONS

While the NETO framework is more general, we focus on the case where $\mathcal{L}$ is a representation learning objective and $R$ denotes reward. This choice gives NETO the potential to explain several recently observed phenomena describing neural representation changes during task learning. We discuss three examples: representation drift, neural reassociation, and the orthogonalization of task-relevant features.

Representation drift describes the slow change in the neural codes of stimuli over time. Crucially, these changes maintain the information encoded in the neural population (Driscoll et al. (2017); Rubin et al. (2015); Ziv et al. (2013)). Representation drift emerges naturally from the NETO framework when $\mathcal{L}$ is a representation learning objective. We saw representation drift clearly in the contextual bandit task: the network responses rotated while maintaining their population structure. This result has been noted before in (Qin et al. (2023)). NETO suggests a potential feature of this drift: It can aid task learning by sampling behavioral policies, even when behavioral performance stabilizes.

In neural reassociation, the brain forms new stimulus-action associations by *reassociating* stimulus-code pairs instead of generating new activity patterns. Researchers observed neural reassociation in primate motor cortex during task learning (Golub et al. (2018)). NETO can reproduce neural reassociation with the correct motor function. Consider a nonlinear motor function $M(\boldsymbol{x})$ from some unspecified brain region to motor cortex (instead of directly to an action). Let the function

be nonlinear so that changing the distribution of inputs does not change the range of $M(\boldsymbol{x})$, which we can think of an as a manifold in activity space. Suppose NETO operates only at the neurons in the domain of $M$. Then, task learning will occur by changing the structure of the neural activity in its domain to re-map stimuli to different but already existing patterns in motor cortex. If we were to record the neural activity in motor cortex during task learning, we would not observe significant changes in the manifold of response patterns. Instead, we would see the associations between these stimuli and the existing patterns shift. NETO can therefore explain neural reassociation.

Nervous systems have also been observed to orthogonalize task-relevent features to facilitate task learning (Failor et al. (2021); Gurnani & Cayco Gajic (2023)). NETO can orthogonalize task-relevant features by exploring the solution space of a nonlinear representation learning objective. As a simple example, consider the non-negative similarity matching (NNSM) objective introduced in (Pehlevan & Chklovskii (2014)). In (Sengupta et al. (2018)), the authors found that networks operating under the NNSM objective learn localized receptive fields. Moreover, the locations of these receptive fields drift under noisy weight updates (Qin et al. (2023)). Drifting receptive fields is an example of feature orthogonalization: Two stimuli can go from sharing a receptive field (large neural code overlap) to existing in different receptive fields (low overlap). This strongly suggests that a NETO network with $\mathcal{L}$ as the NNSM objective would orthogonalize task-relevant features if this improved its ability to seek reward. The observation that NETO may reproduce a number of experimental observations supports it as a biologically plausible theory for emergent policy learning.

## 5.2 LIMITATIONS & FUTURE DIRECTIONS

NETO's primary limitation is that it relies on diffusive searches, which are notoriously inefficient in high dimensions. Consider scaling the agent's representation space to $N$ dimensions. Then suppose that it interacts in a contextual bandit task, this time with $N$ actions and $N$ input clusters. The time that it takes the network to orient these clusters with a diffusive search scales exponentially in $N$.

There are two potential solutions to this issue. The first is a biological argument that downplays it, and the second presents an avenue to address the problem. Biological neural codes tend to be low dimensional (Iyer et al. (2022); Shine et al. (2019); Ebitz & Hayden (2021)). If the diffusive search is largely restricted to this low-dimensional subspace, then it avoids the exponential scaling that comes with an increasingly high-dimensional representation space. This argument, while plausible, avoids the issue. To solve it, we need to construct a more efficient modulatory system. In all our experiments, we assume that the modulatory system acts uniformly on all neurons in the network. This unnecessary assumption restricts the weight diffusion tensor to a multiple of the identity. Neuromodulation in natural systems is not uniform, which may allow it to shape more complex diffusion tensors. It is possible that carefully shaping the diffusion tensor could make policy learning more efficient. This becomes clear when we discretize time into larger episodes, and consider the policy in each episode as a "sample" of the reward function $R$ at a point in weight space. Under this picture, we see that a modulatory system following an evolutionary algorithm like Covariance Matrix Adaptation may be more efficient.

Animals learn tasks quickly by applying previously acquired knowledge to new problems. They achieve this rapid, generalizable learning by leveraging their world model: a powerful predictive representation of the environment and how it evolves under their actions. Future work could also explore whether agents with expressive representation learning objectives that build world models can solve challenging tasks. It would also be interesting to explore other applications of the NETO framework: for instance, online supervised learning.

## 5.3 CONCLUSION

Though we show that a one-layer network can solve basic tasks, the principal contribution of this work is to introduce NETO as a theory for emergent policy learning in nervous systems. This result suggests further exploration into reward-dependent noise in the nervous system.

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

# A APPENDIX

## A.1 NEURAL THERMAL OPTIMIZATION AS A MORE GENERAL FRAMEWORK

While the focus of this work is to introduce NETO for policy optimization, we feel that it is important to briefly introduce the framework in a more general setting. NETO is a framework for how the nervous system can satisfy two distinct objectives when the primary network's dynamics are governed by only one. Fundamentally, NETO states that:

1. Nervous systems are noisy,
2. A self-supervised objective $\mathcal{L}$ determines the network's deterministic weight dynamics,
3. A neuromodulatory system shapes the noise given the value of $R$.

They key insight behind NETO is that if the eigenvalues of the weight diffusion tensor are monotonically decreasing functions of $R$, the attractive points of the system change to balance the optimization of $\mathcal{L}$ and $R$. The only requirement is that $\mathcal{L}$ is differentiable with respect to the network weights. Nicely, $R$ does not have to satisfy this requirement. In a sense, NETO combines a gradient descent algorithm to optimize $\mathcal{L}$ with an evolutionary algorithm to optimize $R$. The evolutionary algorithm comes into play in the design of the weight diffusion tensor.

NETO's generality allows future work to explore other choices of $\mathcal{L}$ or $R$. For instance, if $\mathcal{L}$ were powerful enough to build an expressive world model, then the agent could likely solve challenging reinforcement learning problems. Moreover, if $R$ were a supervised learning loss instead of reward, the agent could learn a classification task online. There are a wealth of different objective combinations that one could consider.

We also note that one could further generalize this framework to any model defined by the parameters $\theta$ that aims to satisfy two objectives $\mathcal{L}$ and $R$, though only $\mathcal{L}$ is differentiable with respect to the model parameters. This framework may be useful if one believes that satisfying $\mathcal{L}$ will help the model approach optimal solutions with respect to $R$, thereby increasing the efficiency of an evolutionary algorithm.

## A.2 NETO EXHIBITS EFFICIENT KNOWLEDGE TRANSFER ACROSS TASKS

Organisms do not relearn the dynamics of their environment each time they face a new task. Instead, they transfer their knowledge from past experiences, using previously learned representations to learn new sequences of actions. NETO agents demonstrate similar knowledge transfer across tasks in online learning settings. Specifically, when a task switch occurs, NETO agents leverage the information already encoded in their representation learning networks to adapt and optimize a new policy.

We illustrate this idea with a modified version of the contextual bandit task considered in Section 4.1. Previously, the agent needed to learn to take the action $a$ corresponding to the random sign

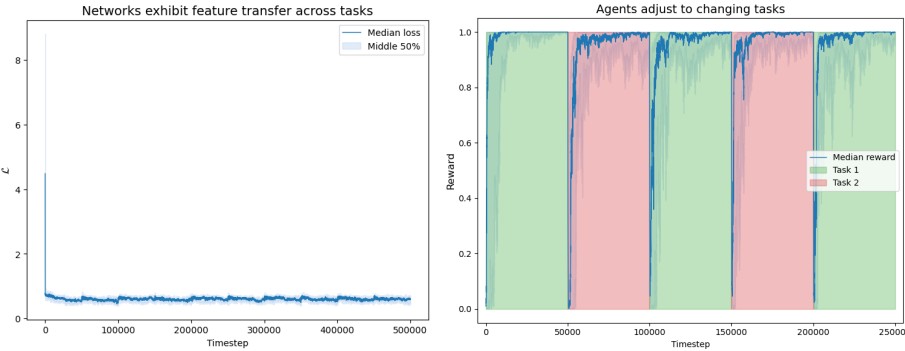

**Figure 6:** *(Left) The principal subspace loss remains low as tasks change, facilitating efficient knowledge transfer. (Right) The agent adjusts readily to changing tasks.*

$k$. In the modified task, the rewarding action-sign association changes from $a = k$ to $a = -k$ on a timescale unknown to the agent. Each association represents a different task. Figure 6 (right) illustrates that the agent adapts to the changing tasks. Importantly, the agent does need to relearn to extract the principal subspace of the inputs (Figure 6, left). Instead, it only needs to relearn how to use this information. Knowledge is transferred efficiently as the tasks evolve.

The NETO agents we focus on in this work quickly optimize their representation learning objective. As a result, transferring learned features between tasks doesn't significantly affect their efficiency. However, for agents with more complex representation learning objectives, this transfer becomes crucial. It allows agents to learn the dynamics of the environment just once, after which they can use this understanding to solve a variety of tasks.

### A.3 The role of activity noise in NETO policy optimization

This work shows that reward-dependent *weight* noise facilitates policy optimization in networks otherwise designed to learn representations. It is natural to ask: Can activity noise play the same role? Or is its only effect to make the agent's policy nondeterministic? Here, we answer this question by showing that activity noise is an "effective" weight noise, and therefore, the network can harness it for policy optimization. Intuitively, activity noise acts as an effective weight noise because the weight updates depend on the activities. We show this in the case of the Similarity Matching objective for concreteness. We consider discrete timesteps throughout. However, it will be clear that the argument applies to any representation learning objective and to continuous time.

Consider the two-neuron network presented in this work. Recall that it has activities $\boldsymbol{x} \in \mathbb{R}^2$ and weights $\theta = (\boldsymbol{W}_0, \boldsymbol{W}_1)$. Let $\boldsymbol{x}_0(\boldsymbol{s})$ denote the network's response to a stimulus $\boldsymbol{s}$ without the presence of noise. Then the network's response with noise $\boldsymbol{x}$ is given by

$$\boldsymbol{x} = \boldsymbol{x}_0(\boldsymbol{s}) + \mathbf{z} \tag{20}$$

where $\mathbf{z} \in \mathbb{R}^2$ is a random variable with covariance matrix $\sigma_x \sigma_x^T$, following equation (1). Per equations (5-6), considering discrete timesteps and setting the weight noise to zero, the weight updates are given by

$$\Delta \boldsymbol{W}_0 = \eta_\theta \left[ (\boldsymbol{x}_0(\boldsymbol{s}) + \mathbf{z}) \boldsymbol{s}^T - \boldsymbol{W}_0 \right] \tag{21}$$

$$\Delta \boldsymbol{W}_1 = \eta_\theta \left[ (\boldsymbol{x}_0(\boldsymbol{s}) + \mathbf{z})(\boldsymbol{x}_0(\boldsymbol{s}) + \mathbf{z})^T - \boldsymbol{W}_1 \right] \tag{22}$$

which can be expanded into

$$\Delta \boldsymbol{W}_0 = \eta_\theta \left[ \boldsymbol{x}_0(\boldsymbol{s}) \boldsymbol{s}^T - \boldsymbol{W}_0 \right] + \eta_\theta \mathbf{z} \boldsymbol{s}^T \tag{23}$$

$$\Delta \boldsymbol{W}_1 = \eta_\theta \left[ \boldsymbol{x}_0(\boldsymbol{s}) \boldsymbol{x}_0(\boldsymbol{s})^T - \boldsymbol{W}_1 \right] + \eta_\theta \left[ \mathbf{z} \boldsymbol{x}_0(\boldsymbol{s})^T + \boldsymbol{x}_0(\boldsymbol{s}) \mathbf{z}^T + \mathbf{z}\mathbf{z}^T \right] \tag{24}$$

We recognize that the first term in each equation corresponds to the weight updates without activity noise. The second terms are effective weight noise terms; they depend on $\mathbf{z}$. As such, NETO policy learning can also occur through activity noise (or some combination of activity and weight noise). Nicely, note that $\eta_\theta$ modulates this effective weight noise as it does the pure weight noise. This is particularly interesting given evidence that dopamine, a neuromodulator associated with reward signals, has been observed to tune the weight learning rate and the activity signal-to-noise ratio in certain regions of mammalian nervous systems (Coddington et al. (2023); Vander Weele et al. (2018); Winterer & Weinberger (2004); Kroener et al. (2009)).

### A.4 The biological plausibility of the NETO noise hypothesis

Here, we discuss the biological plausibility of the NETO reward-dependent noise hypothesis. Is there any evidence that the brain uses reward-dependent noise? Directly measuring weight noise is experimentally challenging because it is difficult to precisely track synaptic weight dynamics. However, we can look for indirect evidence. For example, in this work, the neuromodulatory system tunes the weight noise through the weight learning rate $\eta_\theta$. This aligns with findings showing that

dopamine, a neuromodulator associated with reward signals, adapts the weight learning rate in mice during policy optimization (Coddington et al. (2023)). Additionally, in Appendix A.3, we find that activity noise can serve as an effective weight noise. Research suggests that dopamine tunes the activity signal-to-noise ratio in certain regions of mammalian nervous systems (Vander Weele et al. (2018); Winterer & Weinberger (2004); Kroener et al. (2009)). Together, these examples provide indirect evidence that dopamine could adapt the weight noise in response to reward signals, supporting the NETO reward hypothesis. However, future research is needed to investigate this claim in greater detail.

## A.5  APPROXIMATION OF $R(\phi)$

In this section, we approximate $R(\phi)$ as a function of the contextual bandit state distribution. This calculation is necessary to formulate the agent's learning process as angle-dependent rotational diffusion. Our strategy will be as follows:

1. We will write $a_\phi(\boldsymbol{s})$, the agent's action under an observation $\boldsymbol{s}$ as a function of the angle $\phi$.
2. We will write $R(\phi)$ as an expectation value and eventually an integral.
3. We will see that evaluating this expression requires finding the distribution of the network responses $\boldsymbol{x}(\boldsymbol{s})$. To find this distribution, we compute the principal subspace of $\boldsymbol{s}$.
4. We then use this information to write the distribution of $\boldsymbol{x}$ as a function of $\phi$.
5. Finally, we approximate the integral in step (2).

### A.5.1  STEP 1: FINDING $a_\phi(\mathbf{s})$

We begin by finding $a_\phi(\boldsymbol{s})$. As in the main text, we assume that the agent's parameters are in the solution space of $\mathcal{L}$. It follows that its activity dynamics (4) that its parameters $\theta$ define a mapping from the observation space to the network's representation space given by

$$x(\boldsymbol{s}|\phi) = \boldsymbol{U}(\phi)\boldsymbol{P}\boldsymbol{s} \tag{25}$$

where the operator $P$ projects $\boldsymbol{s}$ into its 2-dimensional principal subspace, and $\boldsymbol{U}(\phi) \in \mathbb{R}^{2\times2}$ rotates the result by $\phi$ radians. By convention, we choose $\boldsymbol{P}$ such that $\boldsymbol{x}(k\boldsymbol{\mu}|\phi = 0) = ck\boldsymbol{v}$ for some $c > 0$.

Since $M(\boldsymbol{x}) = \text{sign}(\boldsymbol{v}^T\boldsymbol{x})$, with $\boldsymbol{v}^T = \frac{1}{\sqrt{2}}(-1, 1)$, this corresponds to the action

$$a_\phi(\boldsymbol{s}) = \text{sign}(\boldsymbol{v}^T\boldsymbol{U}(\phi)\boldsymbol{P}\boldsymbol{s}) \tag{26}$$

### A.5.2  STEP 2: WRITE $R(\phi)$ AS AN INTEGRAL.

Recall that the goal of the agent is to pick the action $k$, the random sign. Then the reward at a single step, $r(\boldsymbol{s}, a)$, is given by

$$r(\boldsymbol{s}, a_\phi(\boldsymbol{s})) = \mathbb{I}\left[a_\phi(\boldsymbol{s}) = k\right] \tag{27}$$

Since the distribution of $\boldsymbol{s}$ is stationary and the agent's policy is fixed, the expected reward $R(\phi) = \mathbb{E}\left[\sum_{t=1}^T r(\boldsymbol{s}_t, a_\phi(\boldsymbol{s}_t))\right] = \mathbb{E}\left[r(\boldsymbol{s}, a_\phi(\boldsymbol{s}))\right]$. So, we can write $R(\phi)$ as

$$R(\phi) = \mathbb{E}\left(\mathbb{I}\left[\text{sign}\left(\boldsymbol{v}^T\boldsymbol{U}(\phi)P\boldsymbol{s}\right) = k\right]\right) \tag{28}$$

$$= P\left[\text{sign}\left(\boldsymbol{v}^T\boldsymbol{U}(\phi)\boldsymbol{P}\boldsymbol{s}\right) = k\right] \tag{29}$$

Let $f_{\boldsymbol{x}}(\boldsymbol{x}_1, \boldsymbol{x}_2|k, \phi)$ be the distribution of the network representations $\boldsymbol{x}$ conditional on the random sign $k$ and given $\phi$. Then, considering the meaning of (24) geometrically, we see that is it equivalent to

$$R(\phi) = \int_{\boldsymbol{x}_1}^{\infty} \int_{-\infty}^{\infty} f_{\boldsymbol{x}}(\boldsymbol{x}_1, \boldsymbol{x}_2|k = 1, \phi) \, d\boldsymbol{x}_1 \, d\boldsymbol{x}_2 \tag{30}$$

We aim to evaluate this integral.

### A.5.3 Step 3: Finding the principal subspace of s

To evaluate the integral (25), we must find the distribution of $\boldsymbol{x} = U(\phi)P\boldsymbol{s}$. Of course, to find this distribution we must first determine $P$. So, we need to find the principal subspace of $\boldsymbol{s}$. Recall that we can decompose $\boldsymbol{s} = k(\sqrt{C}\boldsymbol{z} + \boldsymbol{\mu})$, where $\boldsymbol{z}$ is a standard normal random variable in 10 dimensions, $C$ is the covariance matrix, $\boldsymbol{\mu}$ is the mean, and k is a random sign. Since $\boldsymbol{s}$ is centered (due to the random sign), its covariance matrix is given by

$$\mathbb{E}\left[\mathbf{s}\mathbf{s}^T\right] = \mathbb{E}\left[k^2(\sqrt{C}\mathbf{z} + \boldsymbol{\mu})(\sqrt{C}\mathbf{z} + \boldsymbol{\mu})^T\right] \tag{31}$$

$$= \boldsymbol{\mu}\boldsymbol{\mu}^T + \mathbb{E}\left[\sqrt{C}\mathbf{z}\mathbf{z}^T\sqrt{C}^T\right] \tag{32}$$

where we used the fact that $\mathbb{E}[\mathbf{z}] = 0$. Then, using $\mathbb{E}\left[\mathbf{z}\mathbf{z}^T\right] = I$, and since $C$ is symmetric, $\sqrt{C}\sqrt{C}^T = \sqrt{C}\sqrt{C^T} = \sqrt{C^2} = C$, we find

$$\mathbb{E}\left[\mathbf{s}\mathbf{s}^T\right] = \boldsymbol{\mu}\boldsymbol{\mu}^T + C \tag{33}$$

In the contextual bandit task, we defined $C$ such that $\boldsymbol{\mu}$ is an eigenvector. Let $\lambda_\mu$ be the eigenvalue associated with $\boldsymbol{\mu}$, and $\{\lambda_i\}_{i=1}^9$ denote the other nine eigenvalues with associated eigenvectors $\{\boldsymbol{u}^{(i)}\}_{i=1}^9$. Then the eigenvalues of $\mathbb{E}\left[\mathbf{s}\mathbf{s}^T\right]$ are given by $\{\|\boldsymbol{\mu}\|^2 + \lambda_\mu, \lambda_1, ..., \lambda_9\}$.

The mean vector $\boldsymbol{\mu}$ allows the agent to identify the random sign $k$. So, to ensure that there exists a weight configuration in the solution space of $\mathcal{L}$ that solves the task, we needed the projection of $\boldsymbol{\mu}$ onto the principal subspace of $\boldsymbol{s}$ to be nonzero. We therefore chose $\|\boldsymbol{\mu}\|^2 + \lambda_\mu > \lambda_1 > ... \geq \lambda_9$.

We see that the principal subspace of $\boldsymbol{s}$ is given by $\text{span}(\boldsymbol{\mu}, \boldsymbol{u}^{(1)})$. We can therefore define $P$ as the projection onto this subspace with $P\boldsymbol{\mu} = \|\boldsymbol{\mu}\|\boldsymbol{v}$ and $P\boldsymbol{u}^{(1)} = \frac{1}{\sqrt{2}}(-1, -1)^T$.

### A.5.4 Step 4: Finding the distribution $f_{\mathbf{x}}(\mathbf{x}_1, \mathbf{x}_2 | k = 1, \phi)$

Now that we have determined the projection operator $P$, we can find the distribution of $\boldsymbol{x}$ given the random sign $k = 1$. Notice that, conditioned on $k$, the distribution of $\boldsymbol{s}$ is Multivariate Normal. Any projection or rotation of a Multivariate Normal is still Multivariate Normal. Nicely, this means that the distribution of $\boldsymbol{x}(s) = U(\phi)P\boldsymbol{s}$ is defined by its mean and covariance matrix. We need only compute these.

Recall that $\mathbf{s} = k(\sqrt{C}\mathbf{z} + \boldsymbol{\mu})$, so

$$\mathbb{E}\left[\boldsymbol{x}|k = 1\right] = \mathbb{E}\left[kU(\phi)P(\sqrt{C}\mathbf{z} + \boldsymbol{\mu})|k = 1\right] \tag{34}$$

$$= \mathbb{E}\left[U(\phi)P\sqrt{C}\mathbf{z}|k = 1\right] + \mathbb{E}\left[U(\phi)P\boldsymbol{\mu}|k = 1\right] \tag{35}$$

$$= U(\phi)\|\boldsymbol{\mu}\|\boldsymbol{v} \tag{36}$$

where, in the last line, we used $P\boldsymbol{\mu} = \|\boldsymbol{\mu}\|\boldsymbol{v}$ and $\mathbb{E}[\mathbf{z}] = 0$. Before we evaluate the covariance matrix, we make the following simplification. Let $\boldsymbol{\mu}' := \|\boldsymbol{\mu}\|\boldsymbol{v}$. Notice that

$$\boldsymbol{x}|(k = 1) - \mathbb{E}[\boldsymbol{x}|k = 1] = U(\phi)P(\sqrt{C}\mathbf{z} + \boldsymbol{\mu}) - U(\phi)\boldsymbol{\mu}' \tag{37}$$

$$= U(\phi)P\sqrt{C}\mathbf{z} + U(\phi)P\boldsymbol{\mu} - U(\phi)\boldsymbol{\mu}' \tag{38}$$

$$= U(\phi)P\sqrt{C}\mathbf{z} \tag{39}$$

We can now evaluate the covariance matrix, which we denote $C'$.

$$C' = \mathbb{E}\left[(U(\phi)P\sqrt{C}\mathbf{z})(U(\phi)P\sqrt{C}\mathbf{z})^T\right] \tag{40}$$

$$= U(\phi)P\sqrt{C}\,\mathbb{E}\left[\mathbf{z}\mathbf{z}^T\right]\sqrt{C}^T P^T U(\phi)^T \tag{41}$$

$$= U(\phi)P\sqrt{C}\sqrt{C}^T P^T U(\phi)^T \tag{42}$$

$$= U(\phi)PCP^T U(\phi)^T \tag{43}$$

$$= U(\phi)DU(\phi)^T \tag{44}$$

where $D = \mathrm{diag}(\lambda_\mu, \lambda_1)$. We made the last step by the definition of the projection operator $P$.

This means that

$$\mathbf{x}|(k=1) \sim \mathcal{N}(U(\phi)\mu', U(\phi)DU(\phi)^T) \tag{45}$$

We can now attempt to evaluate the integral (25).

### A.5.5 STEP 5: APPROXIMATING THE INTEGRAL FOR $R(\phi)$

We now see that

$$R(\phi) = \int_{\mathbf{x}_1}^{\infty}\int_{-\infty}^{\infty}\mathcal{N}(\mathbf{x};\ U(\phi)\mu', U(\phi)DU(\phi)^T)\,d\mathbf{x}_1 d\mathbf{x}_2 \tag{46}$$

This integral is not analytically tractable because the Gaussian is not centered and has a potentially non-diagonal covariance matrix. However, we can make an approximation that will be valid for our contextual bandit task.

Notice that we defined the contextual bandit state distribution such that $\lambda_\mu << \|\mu\|^2, \lambda_1$. We did this so that the agent can learn to solve the task consistently (the two Gaussian clusters need to be linearly distinguishable). We therefore approximate the expression with the limit that $\lambda_\mu \to 0$.

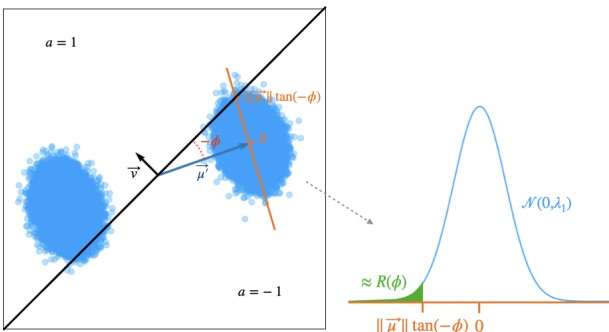

**Figure 7:** *A visualization of the $R(\phi)$ approximation. $R(\phi)$ is equivalent to the probability that $\mathbf{x}(\mathbf{s}|\phi, k=1)$ corresponds to $a = 1$. Visually, this corresponds to the probability that the rightmost cluster is in the upper region of activity space. In the limit that $\lambda_\mu \to 0$, all variance perpendicular to the orange line disappears. So this quantity becomes the integral of the distribution $\mathcal{N}(0, \lambda_1)$ from $-\infty$ to $\|\mu\| \tan(\phi)$.*

For geometric intuition, refer to Figure 6. We see that, in this limit, the integral would be exactly given by

$$R(\phi) = \int_{-\infty}^{\|\boldsymbol{\mu}'\| \tan(\phi)} \mathcal{N}(x; 0, \lambda_1) \, dx \tag{47}$$

$$= \Phi\left(\pm \frac{\|\boldsymbol{\mu}\|}{\sqrt{\lambda_1}} \tan(\phi)\right) \tag{48}$$

since $\|\boldsymbol{\mu}'\| = \|\boldsymbol{\mu}\|$, where we take the negative sign when $\phi \in [\frac{\pi}{2}, \frac{3\pi}{2}]$ and the positive sign otherwise.

### A.5.6 Our approximation agrees with simulations

In Figure 7, we verify our approximation by comparing it to simulations. We find good agreement when our assumption that $\lambda_\mu << \|\boldsymbol{\mu}\|^2, \lambda_1$ holds. To get simulation values of $R(\phi)$, we define the mapping $a_\phi(\boldsymbol{s})$. We then draw the states $\boldsymbol{s}$ according to the process defined in the main text and compute the expected value of the indicator that $a_\phi(\boldsymbol{s}) = k$, where $k$ is the random sign.

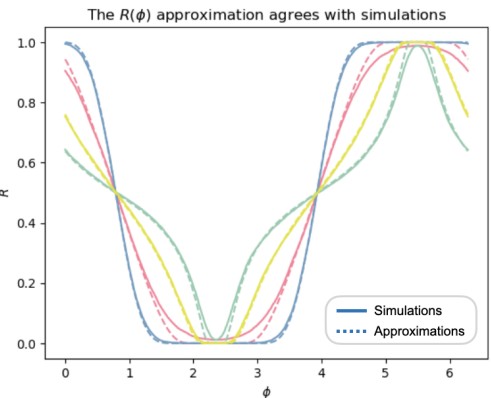

**Figure 8:** *The $R(\phi)$ approximation is reasonable across a range of examples where our assumption $\lambda_\mu << \|\mu\|^2, \lambda_1$ holds. Solid lines indicate simulation values, and dashed lines denote the analytical approximation. In these simulations, $\|\mu\|^2 \in \{0.25, 1\}$, $\lambda_1 \in \{0.1, 2\}$, and $\lambda_\mu = 0.05$.*

### A.6 Optimal Cart Pole Policies do not exist in the solution space of $\mathcal{L}$

Here, we show that optimal Cart Pole policies cannot be found within the solution space of $\mathcal{L}$. We begin with an intuitive argument. Suppose, for the sake of contradiction, that the agent can maintain a successful policy while satisfying $\mathcal{L}$. A successful Cart Pole policy minimizes the angular deviation of the pole from its center. However, because the agent satisfies $\mathcal{L}$, it projects its observations onto their two-dimensional principal subspace. The pole's angular deviation does not lie in this subspace, as its variance is too small. Without access to this critical information, the agent cannot determine which action to take, and thus cannot have an optimal policy. This leads to a contradiction.

It is necessary to validate this intuition because the other observations (cart position, velocity, etc.) should have small variances in an optimal policy as well. We test the claim that an agent with an optimal policy cannot project its observations onto their 2D principal subspace and maintain an optimal policy. We record the observations of an agent with a policy that beats Cart Pole and compute this subspace. We then define a new agent with a motor function $M(\boldsymbol{x}) = \text{sign}(\boldsymbol{v}^T \boldsymbol{x})$ that acts directly on these principal subspace projections. We explore all rotational configurations of the principle subspace projections but find that none result in a policy that beats Cart Pole (Figure 9). An agent cannot project its observations onto their principal subspace if it intends to maintain its

optimal policy[7]. We conclude that agents cannot simultaneously exist in the solution space of $\mathcal{L}$ and have an optimal policy. This result empirically confirms the claim that optimal policies do not exist within the solution space of $\mathcal{L}$.

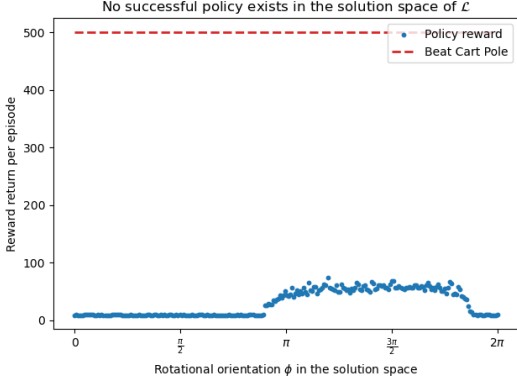

**Figure 9:** *No rotational orientation of the network activities can solve Cart Pole if the observations are projected onto their principal subspace. Points give the median reward return for an agent with a given rotational orientation. $\phi$ is the same angle depicted in Figure 2.*

### A.7  Agent parameters in the two tasks

Table 1 provides the parameters that we used to run the agent simulations in the contextual bandit and Cart Pole tasks. The agent requires greater noise variance to learn features that its representation learning objective does not capture, as seen in the Cart Pole task. However, due to the large variance, we did not add noise to $W_1$. We made this choice because the SM objective requires that $W_1$ remain symmetric. Alternatively, we could have chosen to symmetrize this weight noise.

**Table 1:** *Agent Parameters*

| Task | $\beta$ | $\sigma_\theta^2$ | $\gamma$ |
|---|---|---|---|
| Contextual Bandit | $\{0, 1, 2, 3, 5\}$ | 0.001 | 1 |
| Cart Pole | $\frac{1}{30}$ | 1 | 1 |

### A.8  Implementation of the modulatory system $R$ estimates

In this article, we often referred to a modulatory system that estimates $R$. However, we did not mention how this system implements this policy reward estimation.

#### A.8.1  The contextual bandit

In the contextual bandit task, the modulatory system estimates the reward $R$ with an exponential filter of length $T = 100$ timesteps. It updates its reward estimate at time $t$, $\hat{R}_t$, with the equation

$$\hat{R}_{t+1} = (1 - \frac{1}{T})\hat{R}_t + \frac{1}{T}r(s_t, a_t) \tag{49}$$

where $r(s_t, a_t)$ gives the reward received in that timestep.

---

[7]We assumed here that the result in Figure 9 hold for all successful policies, not just the ones tested. This assumption is reasonable because Cart Pole is a simple task; there are not successful policies that lead to drastically different input distributions.

### A.8.2 CART POLE

In Cart Pole, the network uses the last trial result as the $R$ estimate. This is equivalent to the contextual bandit implementation if we consider the exponential filter to be over trials instead of timesteps, and we set $T = 1$.

