# OpenReview forum: "Policy optimization emerges from noisy representation learning"
_ICLR.cc/2025/Conference — Submitted to ICLR 2025_

### Official Review · Reviewer_DznG · 2024-10-24

**Soundness:** 2
**Presentation:** 3
**Contribution:** 2
**Rating:** 5
**Confidence:** 4

**Summary:**

The authors propose a new algorithm, NETO, for representation learning within the context of Contextual Bandits and Markov Decision Processes.
Their algorithm is posed as a theory of representation learning within the central nervous systems, and as a direct competitor to neuromodulation-based learning rules in e.g. spiking neural networks.

At the core of NETO, both the network activities and the network weights perform noisy gradient descent on a representation learning loss, which they task to be the Similarity Matching objective. Descent on this objective yields Hebbian learning with linear weight decay for both the feedforward and recurrent weights. The noise in the weight update rules is modulated by reward, with higher reward yielding less noise; this allows the network to explore possible representations until one is found for which the resultant policy performs well.

NETO is tested on both a contextual bandit task - corresponding to discrimination of a pair of linearly separable clusters in input space - and the control task Cart Pole. The authors demonstrate by a mathematical analysis that the stationary distribution of solutions learned by NETO has high mass on high-reward solutions, and that as modulation is sharped ($\beta \to \infty$) the stationary distribution sharpens to an exact solution.

The authors then demonstrate that NETO can learn in an MDP setting by testing its performance on Cart Pole. They show that, under a variety of input normalisations, NETO still successfully solves the Cart Pole task.

Lastly, the authors discuss experimental observations that connect with the NETO framework.

**Strengths:**

Quality:
- The theoretical analysis of NETO's behavior for the contextual bandit task is the strongest part of the paper, and is very solid. The mathematical details provide useful and valuable insight into how NETO behaves, and why it is able to successfully solve the task.

Originality:
- The combination of similarity matching objectives with reward modulated noise is novel, and presents a specific, distinct learning rule that a neural network can implement.

Clarity:
- The paper was in general very clearly written, with most arguments easy to follow and engage with. The clarity of the paper was high.

Significance:
- The paper proposes a novel learning rule which can be implemented locally and therefore could be of interest to the neuroscience community.

**Weaknesses:**

The primary weakness of the paper is that it is unclear what problem it is addressing, and whether it wishes to be evaluated as a contribution to computational neuroscience or a contribution to machine learning.

If the paper is intended as a work of computational neuroscience, it has significant shortcomings on multiple accounts:
1. (038) "[NETO] hypothesizes that the noise in the nervous system is reward-dependent, decreasing in magnitude when the nervous system's weight configuration learns a rewarding policy." No direct experimental evidence is given for this claim within the paper, at least not when it is made. The value of this hypothesis would then only be in indirectly explaining other experimental results; but if so, which ones? The hypothesis is presented with minimal indication of why a neuroscientist would accept it as true.
2. (157) In (4), the activity noise is set to zero. Earlier in (037) you write that "NETO treats the brain's intrinsic noise as a feature rather than a bug". Do you mean only noise in weights? If so, you should say this. But then how does NETO relate to noise in activities?
3. (082) If spiking neural networks can learn Atari games (Capone and Paolucci, 2024), can NETO also learn to solve those games? If not, this seems like a major strike against NETO as a plausible theory of representation learning. If so, you should provide evidence. Additionally, you write that previous work with SNNs "was based on rules with little experimental validation". What is the experimental validation that NETO has that these rules do not? For example, what experiment should I look at to see why the form of the weight noise (9) is plausible?
4. (060) A minor detail, but if this is a neuroscience paper then the coloured areas you have in Fig. 1 really ought to correspond to correct brain areas. Modulation should be in the basal ganglia - you have it in occipital/temporal cortex; representation learning should be in occipital cortex/PPC/PFC - you have it in sensory cortex and motor cortex; Motor should be in motor cortex - you have it in the frontal lobe. If your theory is entirely abstract, and you do not intend for modulation to be interpreted as corresponding to the dopaminergic system, you ought not to say in (087) "we return to the idea that a neuromodulatory system acts on a network operating under experimentally validated plasticity rules", as it gives the misleading impression that you intend for your theory to correspond to biology.
5. (480) What falsifiable claims does NETO make about representational drift which could be experimentally tested, and distinguish it from other theories of representation learning? Additionally in (506), you speculate that the behaviour of NETO with the NNSM objective may recapitulate experimental results; but then in (508) seem to imply that this speculation should be taken as experimental support for NETO. If your claim regarding NNSM is correct, please demonstrate so empirically. Otherwise, this speculation does not provide even circumstantial evidence for NETO's validity.


Alternativley, if the paper is intended as a work of machine learning, its empirical results are not strong enough for it to be a contribution of general interest.
1. The only network architecture considered in this work is a 1 layer inhibitory recurrent network without non-linearities, and perceptron-style readout. Does NETO scale to more complicated architectures? In (121), you make the observation that NETO can be *applied to* arbitrary architectures - but this is very different for it *working* for more complicated architectures.
2. The performance of NETO, treated as an ML algorithm, is not benchmarked on either of the tasks it performs. While you show that it can learn to perform a simple contextual bandit task, you do not compare its empirical regret against other learning rules. Similarly, no benchmark is given to understand how NETO's performance on Cart Pole compares to that of, e.g., a TD-learning-style algorithm. These failings seem significant if NETO is to be understood as a competitor to classical TD-learning rules.
3. The tasks that the network performs are simply not complicated enough to be of interest from a machine learning perspective. How does NETO perform on harder tasks, like the continuous control MuJoCo suite or the discrete Atari suite?


Additional weaknesses:
- (309) "Previous work used linear stability analysis to show that the diffusion coefficient characterizing this process is linear in $\eta_\theta$". Please cite the work to which you are referring. Be more explicit in the argument being made here.
- (420) I find the argument made for why projection into the principle subspace is insufficient to solve Cart Pole to be unconvincing. While you are correct that the variance of the pole angle is low, a successful policy will additionally have low variance for the other variables - momentum, angular momentum, and position. It is unclear to me that the pole angle will definitely fall outside of the leading PCs. If your argument is correct, I would like to see a numerical experiment which demonstrates the claim being made, namely that if you train a linear readout on the top 2 PCs of the observations for the Cart Pole optimal policy, you cannot recover a good policy for the task.

**Questions:**

Is the SM objective an intrinstic part of NETO, or do you view it as just one possible choice for the representational objective L? If it is only one choice, can you provide evidence that NETO still functions when another representational objective is used?

---

> ### Author Response · Authors · 2024-11-22
> **Response to Reviewer DznG (Part 1)**
>
> Thank you for your valuable and detailed feedback! We have provided our responses below.
>
> The goal of this paper is to address the question: Can policy optimization be implemented with plasticity rules designed for representation learning? This question is biologically relevant given the mounting evidence that most plasticity in the brain is designed for representation learning. NETO is a new biologically plausible learning algorithm. But more importantly, it introduces the novel concept that the brain could exploit its intrinsic noise to implement policy optimization from a system otherwise designed to extract useful information from the environment.
>
> Neuroscience (part 1)
>
> [Response to comments on the value of the NETO noise hypothesis and the role of activity noise]
> 1. On the roles of activity and weight noise: When we write that “NETO treats the brain's intrinsic noise as a feature rather than a bug,” we mean both the weight and activity noise. We choose to set the activity noise to zero in this work for simplicity, as the activity noise has two conceptually distinct effects on the network. As mentioned in section 3.2.2, the activity noise makes the agent’s policy stochastic because the network’s responses to stimuli are noisy. However, the activity noise also acts as an effective weight noise; the network weights evolve according to the activities, so if there is noise in the activities, this noise will carry through to the weight updates. Therefore, in principle, the brain could take advantage of the weight and activity noise to learn optimal policies (hence the statement in quotes). We focused on the former because the activity noise has the additional effect of creating a stochastic policy, which in principle could be useful but adds unnecessary complexity to the discussions in this work. We appreciate this important point and have added a new section (Appendix A.3) explaining this nuance. We direct the reader to this section when introducing weight noise in line 202.
> 2. On the value of the NETO noise hypothesis: The hypothesis has value on three fronts. First, it proposes a normative role for noise in the brain by showing that a noisy representation learner can also learn to optimize a policy if its noise is modulated. This follows a similar line of reasoning to recent work that makes hypotheses based on normative computational principles, see [1-3], which is a powerful method for understanding neural computation. Second, there is evidence for noise modulation in the brain. Dopamine is known to control the signal-to-noise ratio in (at least) medial prefrontal cortex [4-6], as noted in Appendix A.3 and A.4. Though this typically refers to the activity signal to noise ratio, as previously noted, activity noise is an effective weight noise. Third, the noise hypothesis has the potential to reproduce several experimentally observed phenomena, as discussed in section 5.1. We thank you for this question and added a note on line 207 pointing the reader to Appendix A.4 for a brief discussion on the biological plausibility of noise modulation.
>
> [Response to comments regarding the comparison to other biological models]
> 1. We agree that if more complex NETO agents could not solve Atari games, this would be a major strike against the framework. However, this work does not argue that NETO is an interesting model of learning because it is powerful (though it very possibly is, and future work will explore how far the framework can be taken in terms of task complexity with larger networks that use more powerful representation learning objectives). Rather, it argues that NETO is exciting because it presents a new normative role for noise, and reconciles the empirical evidence that much of the brain’s plasticity is designed for representation learning with the fact that organisms are excellent policy optimizers. Moreover, it achieves this reconciliation without proposing disjoint network modules with different plasticity rules (like Capone et al. 2024 do).
> 2. To your comment about experimental validation of plasticity rules: The SNNs in Capone et al. use two different plasticity rules in two different network modules. In contrast, NETO uses one plasticity rule across the entire network. In the example that we gave, this was the Hebbian rule (derived from the SM objective), which is experimentally validated. In general, NETO can use any number of plasticity rules proposed through computational principles that have seen success in reproducing experimental results (for instance, those in [7]). The specific form of the modulation (9) is an arbitrary choice taken to demonstrate the framework, as stated below the equation. We do not intend to claim that it has been validated experimentally.
>
> (We placed the citations at the end of our multi-part response.)

---

> ### Author Response · Authors · 2024-11-22
> **Response to Reviewer DznG (Part 2)**
>
> Neuroscience (Part 2)
>
> [Response to figure comment]
> 1. Thank you for raising this point. We have edited the figure to reflect the structure of the mammalian brain. However, we would like to comment that this figure is purely schematic. In principle, NETO could be relevant to any organism with a neuromodulatory system. The position of the elements in the mammalian brain is intended only to visually clarify the setup.
>
> [Response to comments on experimental connections]
> 1. We greatly appreciate these comments. To start, NETO is not a theory for representation learning but a theory for how policy optimization can emerge from a representation learner. As such, its predictions do not necessarily compete with the predictions of representation learning theories but rather add to them. The specific formulation of NETO that we discuss was designed to give intuition for the framework and highlight how it learns. It is abstract from a biological perspective because it has two neurons and operates under the SM objective in simple tasks. As such, it is outside of the scope of this work to make specific falsifiable experimental claims regarding representation drift. The purpose of including experimental connections here was to highlight that NETO can qualitatively reproduce several experimentally observed phenomena, and therefore shows promise for a biological theory of learning. The purpose was not to give conclusive evidence or specific predictions, though this will be the subject of future work.
> 2. Regarding your comment about NETO operating under the NNSM objective: this is also a good point. We hesitated to elaborate on the connections to experiment further because we were already short on space. We are working on an experiment to demonstrate our claims empirically. We will incorporate it into the camera-ready version.
>
> Machine Learning
>
> [Response to comment on architectures considered]
> 1. The choice of including recurrent architectures in Figure 1 was intentional but nothing more than suggestive. NETO agents learn rewarding policies because their reward-dependent noise induces a distribution in and around the solution space of a representation learning objective. If a representation learning objective were implemented in a recurrent architecture, mathematically speaking, this same phenomenon would occur, so NETO should still “work.” However, testing a variety of architectures is outside the scope of this work, as the primary purpose is to introduce the NETO framework and associated concepts.
>
> [Response to comment on comparisons with other ML algorithms]
> 1. Since the primary aim of the paper is to introduce NETO as a framework for learning in biological systems, the relevant comparative baselines are the components and assumptions of other biological models, not the learning efficiencies of other algorithms in these simple tasks. NETO is not to be understood as a competitor to classical TD-learning-style algorithms in terms of efficiency.
>
> [Response to comment on task difficulty]
> 1. We completely agree. We reiterate the point that the tasks considered in this work are intentionally simple. This choice ensures that the learning process is amenable to mathematical analysis and therefore easy to understand conceptually. This attribute is more important for this work than testing NETO in complex tasks, given that the purpose of the paper is to introduce the framework as a model of learning in biological systems.
>
> [Responses to “Additional weaknesses”]
> 1. We are grateful that you pointed out the missing citation here. We have corrected this and rephrased our description of the strategy used by the authors. We refrain from being more explicit in their argument because it is quite involved (please see the supplementary information in Qin et al. 2023).
> 2. Thank you for your concern and insightful suggestion. Empirically, we observe that our argument holds only for certain input normalizations. The main goal of the Cart Pole task is to provide an example where optimal policies do not exist within the solution space of $\mathcal{L}$. As such, we've decided to exclude the normalizations where optimal policies can be found within this space. Instead, we only include the example with no normalization. Additionally, we've included the experiment you suggested in a new section (Appendix A.6) and direct the readers there on line 425. With these adjustments, Cart Pole more clearly serves as an example of a task that the agent can only solve by identifying task-relevant features. It also removes the possible confusion around the meaning of normalizations.

---

> ### Author Response · Authors · 2024-11-22
> **Response to Reviewer DznG (Part 3)**
>
> [Response to final question]
> 1. We view the SM objective as only one possible choice of representation learning objective. Crucially, the framework is intended to be more general and incorporate objectives that can build much more powerful representations. The reason that we chose to introduce NETO with the SM objective in great depth (as opposed to testing it on multiple objectives) is that this gives good intuition for why and how NETO agents learn optimal policies: they diffusively search near the solution space of their representation learning objective. With this intuition, our opinion is that it is clear the framework will work with other representation learning objectives. The one caveat, alluded to in section 4.2, is that NETO agents are most likely to find reasonable policies if these policies exist near the solution space of their representation learning objective. (This is perfectly reasonable biologically, since nervous systems must extract “useful” information from the environment in order for organisms to survive.) As long as the objective extracts information suitable for the task, our analysis will generalize. The experiment demonstrating our claims about NETO agents using the NNSM objective will be included in the camera-ready version and will also further support this claim.
>
> Citations:
>
> [1] Lipshutz, David, et al. "Normative framework for deriving neural networks with multicompartmental neurons and non-Hebbian plasticity." PRX Life 1.1 (2023): 013008.
>
> [2] Duong, Lyndon, et al. "Adaptive whitening with fast gain modulation and slow synaptic plasticity." Advances in Neural Information Processing Systems 36 (2024).
>
> [3] Pehlevan, Cengiz, Tao Hu, and Dmitri B. Chklovskii. "A hebbian/anti-hebbian neural network for linear subspace learning: A derivation from multidimensional scaling of streaming data." Neural computation 27.7 (2015): 1461-1495.
>
> [4] Kroener, Sven, et al. "Dopamine modulates persistent synaptic activity and enhances the signal-to-noise ratio in the prefrontal cortex." PloS one 4.8 (2009): e6507
>
> [5] Winterer, Georg, and Daniel R. Weinberger. "Genes, dopamine and cortical signal-to-noise ratio in schizophrenia." Trends in neurosciences 27.11 (2004): 683-690.
>
> [6] Vander Weele, Caitlin M., et al. "Dopamine enhances signal-to-noise ratio in cortical-brainstem encoding of aversive stimuli." Nature 563.7731 (2018): 397-401.
>
> [7] Halvagal, Manu Srinath, and Friedemann Zenke. "The combination of Hebbian and predictive plasticity learns invariant object representations in deep sensory networks." Nature neuroscience 26.11 (2023): 1906-1915.

---

> > ### Comment · Reviewer_DznG · 2024-11-26
> > **Clarification appreciated; results are still not strong enough.**
> >
> > I appreciate the long and detailed reply by the authors. I think the changes made by the authors have improved the paper, and as such I am increasing my score. In particular, appendix A.3 - addressing the connection between weight noise and activity noise - was illuminating, and helped me understand why the authors focus only on weight noise in their analysis. Furthermore, the experiment in appendix A.6 is an excellent example of a previously intuitive argument being made rigorous with an experiment.
> >
> > However, I still think that the evidence provided in the paper is insufficient for the strength of the claims made by the authors.
> > Throughout the paper and the rebuttal, the authors maintain that NETO is a **general** framework. For example, they claim that the choice of reward-scaling relationship is arbitrary, and that any monotonically decreasing function will do. However the power of NETO as a general framework is not demonstrated. What is demonstrated is that for one specific choice of implementation, NETO is capable of learning a pair of reasonably trivial tasks. The additions to the paper, at present, are not substantial enough for me to increase past the acceptance threshold.
> >
> > The authors write that the "primary purpose is to introduce the NETO framework and associated concepts." I do not believe that currently the framework is conceptually complex enough or empirically interesting enough to have an entire paper dedicated only to explaining the core concepts. To be clear, I like the idea and find it interesting. But this is true of lots of ideas that do not get published at ICLR. The authors do not do enough to demonstrate why the NETO framework in particular is worthy of our attention. If NETO was an established theory with a string of impressive biological predictions OR numerical results OR mathematical analyses, a paper can reasonably be dedicated entirely to its exposition and clarification. But the current paper puts the cart before the horse; it is overly concerned with a mathematical analysis of a very specific case without doing enough to justify why we should take NETO seriously.
> >
> > ## What I would need to increase past acceptance threshold
> > I would like to see this paper improved to the point that I can recommend it for acceptance. As such I will give a number of concrete suggestions for how the authors can get me to raise my score past a 5.
> >
> > ### A more thorough demonstration of the generality of the framework
> > If the authors wish to maintain that NETO is a general framework they must do more to demonstrate this. I see two plausible approaches to doing so, and I would be content with either:
> > 1. **Diverse experimental results**. If the authors demonstrate that - for a wide range of choices of representational learning objective $\mathcal{L}$ and reward mapping $\eta_\theta(R)$ - a NETO agent can learn a simple task (say CartPole), I would feel like their claims were justified. The authors say that they will include results for NNSM, but this is only one alternative choice. The authors do not do so because "testing a variety of architectures is outside the scope of this work" - but as explained above, without stronger results, the current scope of the work does not merit publication at ICLR.
> > 2. **A stronger and more general mathematical analysis**. The mathematical analysis in the paper is for only one choice of objective and reward mapping. If the authors can show that, under some general family of objectives and reward mappings which satisfy some mild constraints, NETO results in learning, this would also justify their claims.
> > To be clear, I am not requesting the authors do both. *Some* additional arguments are needed to support their claims, and either one of thse paths is a valid strategy.
> >
> > ### Better experimental results *OR* connection with biology
> > As above, I will give two concrete ways that the paper can be strengthened. Either would be fine with me; I am not requesting both.
> > 1. The authors still have no baseline for their learning results. While I understand that this is not a pure ML submission, and so NETO should not be understood as a straightforward TD competator, the lack of *any* baseline against which to compare NETO's performance is still a weakness. There are other frontier theories of reward-modulated biologically-plausible learning. The authors did not in their initial rebuttal demonstrate that NETO compares favourably to these theories in any meaningful sense. Such results would provide reason to take NETO seriously as a competator theory.
> > 2.  Alternatively, the authors can offer a concrete, falsifiable prediction that their theory makes. The authors have clarified that they intend for this to be a contribution to neuroscience rather than ML. I am happy with that, but if so the authors need to take seriously the notion that they are building a scientific theory which could in principle be supported or falsified by empirical evidence.

---

### Official Review · Reviewer_ZG6t · 2024-11-04

**Soundness:** 3
**Presentation:** 3
**Contribution:** 2
**Rating:** 5
**Confidence:** 3

**Summary:**

The paper introduces a novel framework, termed NETO, designed to facilitate policy optimization through a noisy representation learning objective. The efficacy of the framework is demonstrated using two simple RL tasks. Additionally, the paper discusses the biological plausibility of the proposed framework.

**Strengths:**

- The NETO framework seems like a novel contribution that hypotheizes that noisy representation learning can also facilitate policy optimization with reward-modulated noise.
- The paper presents a clear argument for why biological learning could be using a NETO-like framework.
- The paper is well presented and easy to understand.

**Weaknesses:**

- The tasks considered in the paper are very basic to gain any insights into whether this framework will work for more complex settings. While the section on the contextual-bandits task is clear and legible, it feels over-explained as of now. Although the section is helpful in visualizing the diffusive search that the reward-dependent noise induces, much of the explanations feel redundant and can be moved to the appendix.
- Even though the framework is about online agents, the tasks used to evaluate the framework are simple and don't require online-learning. The authors should include tasks that are more suitable for online learning (which can also be a toy task). A simple experiment showing that the framework enables efficient transfer across tasks will also be interesting.
- There is no comparative baseline in the paper to highlight the strengths of the framework.
- If the aim of the paper is to ***only*** serve as an introduction to a biologically plausible framework for policy learning, then a heavy neuroscience focused conference or journal might be a better avenue.

**Questions:**

Suggestions:
- Figure 3 & 5 (left) might look better with log-log scale or log x-scale.
- The x-axis on figure 4 (left) could be more legible with pi-based labels. Eg., labels of $0, \pi/2, \pi, 3\pi/2, 2\pi$.

---

> ### Author Response · Authors · 2024-11-22
> **Response to Reviewer ZG6t**
>
> Thank you for your valuable feedback! We have provided our responses below.
>
> [Responses to comments on task choice]
> 1. On task simplicity: There is a trade-off between analytical tractability and the complexity of the task. Here, we deliberately consider simple tasks in order to analytically follow and understand the learning dynamics. We make this choice because the primary contribution of this work is to introduce the idea behind NETO: Reward-dependent noise can facilitate policy optimization in networks otherwise designed for representation learning. We agree that in future work it will be important to test more complex tasks.
> 2. This is also an insightful point. We chose to focus on tasks that do not require online learning because these tasks are more amenable to analysis, which allows us to highlight how NETO agents learn. However, we appreciate the observation and have added a new section to the Appendix (A.2) that extends the contextual bandit by making the context associations switch on some timescale, which means that the agent’s policy must adapt in real time to changing goals (which is only possible through online learning). We see efficient transfer across tasks; the network does not have to relearn to extract the principal subspace of the inputs, it only needs to relearn how to use this information.
>
> [Response to comment on comparative baselines]
> 1. Since the primary aim of the paper is to introduce NETO as a framework for learning in biological systems, the relevant comparative baselines are the components and assumptions of other biological models, not their learning efficiencies in linear, two-neuron networks. In the Related Work section, we highlight that NETO is more promising than other “baseline” models given the growing consensus that the brain is primarily a network-level representation learning system. To our knowledge, NETO is the only model to date that can solve nontrivial tasks with network plasticity rules that operate primarily on this principle. Moreover, because of its generality, NETO can adopt the previous successes of plasticity rules derived from representation learning objectives in reproducing neural data.
>
> [Response to comment on suitability for ICLR]
> 1. The aim of this paper is primarily to serve as an introduction to a biologically plausible learning framework. However, the ideas presented in this work could also be of interest for pure machine learning applications in the context of efficient online learning. NETO agents can transfer learned representations across tasks. We demonstrate this idea in Appendix A.2. As tasks change, the agent does not have to relearn to extract the principal subspace of its inputs. Instead, it only needs to relearn how to use this information. While this is a simple example, the concept generalizes to agents with more complex representation learning objectives. For them, this information transfer is crucial because it allows the agents to learn the dynamics of their environment just once, after which they can use this understanding to solve a variety of tasks. Additionally, we note that ICLR has a history of publishing works in theoretical neuroscience. For instance, please see:
>     - “A Unified Theory of Early Visual Representations from Retina to Cortex through Anatomically Constrained Deep CNNs” (ICLR 2019)
>     -  “Relating transformers to models and neural representations of the hippocampal formation” (ICLR 2022)
>     - “Disentanglement with Biological Constraints: A Theory of Functional Cell Types” (ICLR 2023)
>     - “Hebbian Deep Learning Without Feedback” (ICLR 2023)
>
> [Responses to suggestions]
> 1. Thank you for this suggestion – we did try log scales but unfortunately it did not make the figures more visually appealing.
> 2. Thank you for this suggestion too. We agree and have edited Figure 4 to use pi-based labels.

---

> > ### Comment · Reviewer_ZG6t · 2024-11-29
> >
> > Thank you for the detailed response and new experiments. The appendix A.2 is insightful in showing that the agents under this framwork quickly adapts to a change in the reward function by utlizing. The new appendices A.3 and A.6 are also helpful in justifying some intuitive claims/assumptions made in the main text.
> >
> > However, I still think that the current version of the paper doesn't resolve the major criticism of the simple task choices and no comparative baseline. As mentioned by other reviewers and myself, the simplicity of the tasks considered doesn't justify the claim that the NETO is a general framework. I like the detailed mathematical analysis and explanation of learning dynamics of the simple tasks in the paper. But I disagree with the justification of only doing the simple task because of analytical tractability. You can analytically explain how the different components of the framework help on the simple task setting, **and** also provide results on more complex tasks without providing much analytical analysis on the complex tasks.
> >
> > Also, since the authors intend this paper to introduce a general framework for learning in biological systems, it would be helpful to specify more concrete research directions or predictions to validate this framework in the future work section.
> >
> > I'm keeping my current score.

---

### Official Review · Reviewer_vJYN · 2024-11-04

**Soundness:** 3
**Presentation:** 2
**Contribution:** 2
**Rating:** 5
**Confidence:** 2

**Summary:**

This paper proposes NEural Thermal Optimization (NETO)—a representation learning method that implicitly learns rewarding policies from reward-dependent noise. The theoretical analysis demonstrates that network activities converge to the solution space of the representation objective, subsequently drifting within this space due to reward-dependent noise, effectively searching for reward-optimizing policies. Intuitively, this is achieved by reducing the noise-scale, and hence the exploration, when the reward increases. Experimental validation is provided through Contextual Bandit and CartPole problems.

**Strengths:**

The overall idea of the paper is interesting and the theoretical analysis is comprehensive and thorough.

**Weaknesses:**

I believe the paper would greatly benefit from using lemmas and theorems to highlight its key contributions. As it stands, it is challenging to parse and distinguish between the rigorous claims the authors make and their intuitive insights.

Additionally, the paper would be strengthened by offering more high-level intuitions before delving into the maths. For example, the idea of "converging to the desired solution space and then exploring it using reward-dependent noise" is intuitively accessible and could be presented earlier in the paper. Currently, for readers who are not very familiar with the related work, this idea only becomes clear after some effort in digesting the math.

Besides that, I have some reservations about the theoretical contributions.

1. What are the guarantees for learning optimal policies? Authors claim "With the appropriate choice of $\eta_{\theta}(R),$ the agent is guaranteed to learn a policy that maximizes reward." (L. 375). What is appropriate choice?
2. What is the applicability of the results? Do they apply only to contextual bandits? Only to contextual bandits with the specific set of parameters?

**Questions:**

1. Why do the authors use MDPs, that are discrete-time with PDEs that imply continuous time?
2. While Similarity Matching is never explicitly defined. I think it would be appropriate to include the definition into the paper.
3. How would your method perform if there are isolated locally optimal policies? Would the diffusive search be able to escape such optima?
4. Figure 5 is missing a legend.

Potentially, I could have missunderstood some claims of the paper. I am willing to increase my score if the authors clarify the confusions and provide explanations to the points above.

---

> ### Author Response · Authors · 2024-11-22
> **Response to Reviewer vJYN**
>
> Thank you for your valuable feedback! We have provided our responses below.
>
> [Responses to comments on clarity]
> 1. There were a range of responses regarding the clarity of the manuscript, from very clear (reviewer DznG) to overly pedantic (reviewer ZG6t) to slightly unclear (reviewer vJYN). We agree that reformatting the paper in terms of theorems and lemmas would highlight the theoretical results. However, this choice would also falsely suggest that the primary contribution of the paper is the results in the contextual bandit task. Rather, the primary contribution is the novel, biologically relevant idea that reward-dependent noise can implement policy optimization in networks otherwise designed for representation learning. This idea is much more general than the two tasks (and even the representation learning objective used), so we hesitate to present results regarding these specific examples through theorems and lemmas. The tasks presented serve to illustrate the main conceptual ideas regarding how NETO agents learn policies.
> 2. With that being said, we greatly appreciate your suggestion to offer more high-level intuition before delving into the calculations and have added this intuition to lines 229-230 before delving into the math.
>
> [Responses to comments on theoretical contributions]
> 1. In general, any choice of $\eta_{\theta}(R)$ that monotonically decreases as a function of $R$ is appropriate, consistent with the requirement outlined in Section 3.2.3 that the noise decrease monotonically as a function of $R$. We focused on the example $\eta_{\theta}(R) = e^{-\beta R}$ for clarity, as the purpose of the contextual bandit task is to illustrate the more general concept that modulating noise facilitates policy optimization. In this example, the agent is guaranteed to learn an optimal policy as $\beta \to \infty$ (though in practice, $\beta \approx 5$ works nearly perfectly, as seen in Figure 4). Therefore, the statement on line 376 was referring to the choice of the parameter beta: It should be large. We replaced the phrase “with the appropriate choice” with the phrase “with strong modulation” to emphasize the fact that we are referring to the choice of beta here. However, in general, any monotonically decreasing function of R is appropriate, with more sharply decreasing functions (compared to the reward scale) leading to more bias towards rewarding policies.
> 2. The specific analysis in Section 4.1 applies only to agents using the similarity matching objective and in the specified contextual bandit task. The NETO framework, however, is completely general in terms of both the representation learning objective used and the task. The examples provided are meant to illustrate how NETO agents learn in scenarios where their learning is analytically tractable and therefore easily understood.
>
> [Responses to additional questions]
> 1. We use PDEs that imply continuous time because NETO is meant as a model for learning in biological systems, which live in continuous time. However, to investigate the model's properties computationally, one must use discrete-time, which makes MDPs a natural choice.
> 2. We included a definition of the similarity matching objective as a footnote in the revised manuscript [line 160].
> 3. This is also a great question. We can look at the contextual bandit task for some intuition. The matter of whether a diffusive search can escape a local optimum is a question about a specific learning trajectory. As we saw in the contextual bandit task, the most natural way to think about learning in the NETO framework is not to focus on a specific trajectory but to consider the limiting distribution over network parameters. This limiting distribution is defined jointly by the representation learning objective and the noise modulation. If there were isolated locally optimal policies, one would find the distribution peaked at each, with larger peaks corresponding to more rewarding policies. As a result, NETO agents are generally more likely to find global optima than to get stuck in local optima. The specific choice of  $\eta_{\theta}(R)$ will determine the relative magnitude of these peaks and will be the subject of future work. However, if one does wish to focus on a specific learning trajectory, the answer is that the agent can escape local optima, though the escape time likely depends nontrivially on the task, the form of noise modulation, and the representation learning objective.
> 4. Thank you for pointing this out as well. We feel that the distinction between different normalizations was a confusing detail and not necessary to the main point that the agent learns task-relevant features not captured by $\mathcal{L}$ to solve Cart Pole. As such, we removed all normalizations. The agent now receives input from Cart Pole without alteration. We have edited Figure 5 and Section 4.2 to reflect this change.

---

### Meta-Review · Area_Chair_buA4 · 2024-12-26

**Metareview:**

This paper introduces a framework that proposes how nervous systems can learn behavioral policies through noisy representation learning. The key idea is that reward-dependent noise in network weights can facilitate policy optimization in networks primarily designed for representation learning. The authors try to demonstrate this through theoretical analysis on a contextual bandit task and empirically on the cartpole environment. The framework suggests that policy optimization can emerge naturally from representation learning when combined with reward-modulated noise.

Positive:
- new framework connecting representation learning and policy optimization
- Biologically plausible mechanism that aligns with experimental observations

Weaknesses:

- limited empirical validation on simple tasks only (contextual bandits and CartPole)
- lack of comparative baselines against other biological learning theories
- unclear positioning between computational neuroscience and machine learning contributions
- no concrete, falsifiable experimental predictions

The paper presents an interesting theoretical framework but fails to provide sufficient evidence to support its claims. While the mathematical analysis is sound, the empirical validation is limited to simple tasks. The paper's positioning between neuroscience and ml leaves both aspects insufficiently developed - neither providing strong ml results nor making concrete neuroscientific predictions.

**Additional Comments On Reviewer Discussion:**

The rebuttal period focused on three main concerns:

- reviewers questioned the simple nature of tasks. Authors justified this choice for analytical tractability but added new experiments in appendices showing online learning capabilities.
- concerns were raised about the generalizability claims. Authors argued for the framework's theoretical generality but provided limited additional empirical evidence.
- questions arose about experimental validation and predictions. authors clarified their neuroscience connections and added discussions on noise modulation in appendices.

while the authors were responsive and made additions to address concerns, the fundamental limitations regarding empirical validation and concrete predictions remained largely unaddressed.

---

### Decision · Program_Chairs · 2025-01-22

Reject